# Epithelial-to-Mesenchymal Transition Is Not a Major Modulating Factor in the Cytotoxic Response to Natural Products in Cancer Cell Lines

**DOI:** 10.3390/molecules26195858

**Published:** 2021-09-27

**Authors:** Baris Kucukkaraduman, Ekin Gokce Cicek, Muhammad Waqas Akbar, Secil Demirkol Canli, Burcak Vural, Ali Osmay Gure

**Affiliations:** 1Department of Molecular Biology and Genetics, Bilkent University, Ankara 06800, Turkey; barisk@bilkent.edu.tr (B.K.); ekincicek@gmail.com (E.G.C.); muhammadakbar@bilkent.edu.tr (M.W.A.); 2Molecular Pathology Application and Research Center, Hacettepe University, Ankara 06100, Turkey; secil.demirkol@bilkent.edu.tr; 3Department of Genetics, Aziz Sancar Institute of Experimental Medicine, Istanbul University, Istanbul 34093, Turkey; vburcak@istanbul.edu.tr; 4Department of Medical Biology, Acibadem University, Istanbul 34684, Turkey

**Keywords:** natural products, cancer, chemotherapy resistance, epithelial-to-mesenchymal transition, mesenchymal-to-epithelial transition

## Abstract

Numerous natural products exhibit antiproliferative activity against cancer cells by modulating various biological pathways. In this study, we investigated the potential use of eight natural compounds (apigenin, curcumin, epigallocatechin gallate, fisetin, forskolin, procyanidin B2, resveratrol, urolithin A) and two repurposed agents (fulvestrant and metformin) as chemotherapy enhancers and mesenchymal-to-epithelial (MET) inducers of cancer cells. Screening of these compounds in various colon, breast, and pancreatic cancer cell lines revealed anti-cancer activity for all compounds, with curcumin being the most effective among these in all cell lines. Although some of the natural products were able to induce MET in some cancer cell lines, the MET induction was not related to increased synergy with either 5-FU, irinotecan, gemcitabine, or gefitinib. When synergy was observed, for example with curcumin and irinotecan, this was unrelated to MET induction, as assessed by changes in E-cadherin and vimentin expression. Our results show that MET induction is compound and cell line specific, and that MET is not necessarily related to enhanced chemosensitivity.

## 1. Introduction

Cancer is the second leading cause of death worldwide and about one in six deaths are due to cancer [1]. Chemotherapy is a main treatment option for cancer and has been used as a monotherapy, or in combination with surgery or radiation therapy, for decades. However, drug resistance, innate or acquired, is a major determinant of treatment response and disease progression. Many studies have shown that epithelial-to-mesenchymal transition (EMT) plays an important role in chemotherapy resistance [2,3,4,5]. Therefore, targeting EMT to improve the response to chemotherapy has recently become a priority in cancer research [6].

Nutrition and diet are effective in reducing cancer risk. Numerous dietary natural products have shown potential in prevention and/or treatment of cancers [7,8,9]. Accumulating evidence shows that naturally occurring bioactive compounds interfere with carcinogenesis by inhibiting tumor cell proliferation and targeting multiple abnormally activated signaling pathways [10,11]. Furthermore, many natural products have been shown to reverse EMT [12,13], and to overcome drug resistance in cancer [14,15].

In line with these observations, modulators of EMT, combined with chemotherapeutics and/or targeted drugs, may possibly improve clinical outcomes in cancer. Most studies aiming to address this issue, however, were conducted either in single cell lines, or only in one cancer type [12]. Hence, a comprehensive study, which evaluates the effectiveness of naturally occurring bioactive molecules in the modulation of EMT and their role as chemotherapy enhancers, is needed. To fulfill this need, we investigated the effects of eight natural compounds: apigenin, curcumin, epigallocatechin gallate (EGCG), fisetin, forskolin, procyanidin B2, resveratrol, urolithin A, and two repurposed agents, fulvestrant and metformin, on cancer cell viability, modulation of EMT phenotype, and sensitization to chemotherapeutics in vitro in three different cancer types. Curcumin is one of the best studied natural products, present in turmeric. Many studies have showed that curcumin sensitizes cancer cells to chemotherapeutics by modulating EMT [16,17,18,19]. Resveratrol (phytoalexin) is a polyphenol, found in grapes, chocolate, and peanuts [20]. The regulatory role of resveratrol in EMT, leading to chemosensitization, has also been extensively studied [15,21,22,23]. Recently, apigenin, a natural flavonoid, was shown to act as an inhibitor of Snail, which is a key regulator of EMT [24,25]. Apigenin inhibits IL-6 linked downstream pathways, resulting in epithelial marker E-cadherin upregulation, and downregulation of the mesenchymal marker N-cadherin [26]. Another natural product, EGCG (epigallocatechin gallate), the most prevalent tea polyphenol, was also shown to reverse EMT, and synergistic effects of EGCG were shown in pancreatic cancer cells when combined with gemcitabine through the modulation of EMT markers [27]. Fisetin, a natural flavanol found in various fruits and vegetables, was shown to reverse EMT in vivo in a xenograft model of the MDA-MB-231 breast cancer cell line [28]. Forskolin, a natural product derived from the root of the plant *Cleus forskohlii*, was shown to induce tumor-initiating cells to undergo mesenchymal-to-epithelial transition, and to cause them to lose their tumor-initiating ability [29]. Procyanidin B2, an abundant polyphenol in red wine, was shown to reverse high-glucose-induced-EMT in a human renal epithelial cell line [30]. Additionally, urolithin A, a metabolite of ellagitannins, was shown to inhibit metastasis in a SW620 colon cancer cell line, and to increase sensitivity against 5-FU in a Caco-2 colon cancer cell line [31,32]. It was also shown to upregulate the epithelial marker E-cadherin and to downregulate the mesenchymal markers, vimentin and N-cadherin, in lung cancer cells [33]. Fulvestrant and metformin are FDA-approved compounds that have been shown to be capable of reducing the mesenchymal features of cancer cells. Fulvestrant is an estrogen antagonist which was shown to reduce the mesenchymal features of lung carcinoma cells, resulting in tumor sensitization to chemotherapy. The anti-diabetic drug metformin was also shown to have capability in the modulation of EMT [34,35,36,37]. These compounds have been reported for their low-toxicity against normal cells in many studies [38,39,40,41,42,43,44,45,46,47,48,49,50,51,52,53]. Here, we report on the activity of eight natural products, and two repurposed agents, in sixteen cell lines of colon, breast, and pancreatic cancer origin.

## 2. Results

### 2.1. Screening of Natural Products in Various Breast, Colon, and Pancreatic Cancer Cell Lines Reveals Notable Anti-Cancer Activity

We first sought to characterize the inhibitory effect of natural products and repurposed agents on various breast, colon, and pancreatic cancer cell lines. We selected eight naturally occurring molecules, an estrogen antagonist, and an anti-diabetic drug that were previously shown to inhibit and/or reverse EMT. The cells were designated resistant if an IC50 value was not observed within the concentration ranges tested. For the breast cancer cell lines, curcumin exhibited a remarkable inhibition of cell viability, with IC50 values ranging from 12 to 40 μM among the cell lines tested (Table 1). Additionally, apigenin and resveratrol showed variance in the inhibitory activity on cell viability, whereas fisetin and forskolin exhibited a more restricted cytotoxicity on cell lines. MDA-MB-453 was the only breast cancer cell line sensitive to fulvestrant, which may be a result of HER2-overexpression in this cell line, while other cell lines were triple-negative. On the other hand, EGCG, procyanidin B2, urolithin A, and metformin, in micromolar concentrations, did not have any effect on cell viability in breast cancer cell lines at the concentrations tested.

Since diet is an important factor in colon cancer risk, studying the effects of natural products on colon cancer may provide new insights into cancer prevention and treatment [54]. In a similar way to our findings in breast cancer, curcumin was the most effective compound for inhibition of viability in colon cancer cell lines, with IC50 values ranging from 6 to 15 μM (Table 2). We observed moderate inhibition of cell viability with apigenin, EGCG, fisetin, resveratrol, and urolithin A, at micromolar levels. In addition, metformin in the millimolar range was cytotoxic for colon cancer cell lines (Table 2). The fact that KM12 is the only cancer cell line sensitive to forskolin among our colon cancer cell line panel, is also evident in the NCI-60 drug sensitivity data (Appendix A) [55]. On the other hand, procyanidin B2 and fulvestrant did not exhibit any inhibitory effects on colon cancer cell viability.

Pancreatic ductal adenocarcinoma (PDAC) is among the most lethal cancer types, and a number of naturally occurring molecules in preclinical studies exhibit promising pharmacological features [56,57]. We examined the effects of natural compounds on the viability of four pancreatic cancer cell lines (Table 3). Similar to what we observed in breast and colon cancer cells, curcumin was the most effective natural compound in terms of pancreatic cancer cell viability. We observed moderate cytotoxicity with apigenin, EGCG, fisetin, and resveratrol. In addition, forskolin and urolithin A exhibited selective and mild effects on the pancreatic cancer cell lines. Millimolar levels of metformin also had growth inhibitory effects in these cells. In a similar way to the colon cancer cell lines, procyanidin B2 and fulvestrant had no effect on cell viability.

In total, 12, 13, and 15 cancer cell lines (out of 16) were resistant (IC50 could not be determined within the concentration ranges tested) to forskolin, procyanidin B2, and fulvestrant, respectively, suggesting that these compounds were not effective inhibitors of cancer cell viability. Apigenin, curcumin, and resveratrol were the only three compounds for which IC50 values could be generated for all of the cell line types tested. Among these compounds, the lowest mean IC50 was obtained for curcumin for all three cancer types, with values of 23.8 μM, 10 μM, and 15 μM for breast, colon, and pancreatic cancer cells, respectively.

### 2.2. Natural Products Induce Mesenchymal-to-Epithelial Transition Selectively in Cancer Cell Lines

We treated six cell lines from three cancer types with natural compounds and evaluated changes in the epithelial/mesenchymal phenotypes (Figure 1 and Appendix A). We used four colon cancer cell lines (Caco-2, KM12, LoVo, and WiDr), among which Caco-2 is known for its high plasticity, with the flexibility to shift through the spectrum between epithelial and mesenchymal phenotypes [58]. Three other cell lines showed variance in expression of EMT marker genes in which KM12 was the most mesenchymal and WiDr was the most epithelial (Appendix A). We performed a 48-h treatment on these cells with the compounds at multiple doses and quantified the change, via qRT-PCR, in the expression of E-cadherin (CDH1) and vimentin (VIM), as markers of epithelial and mesenchymal phenotypes, respectively. An increase in CDH1, and a decrease in VIM expression upon treatment, compared to the untreated control cells, were considered as indicators of MET induction, and vice versa for EMT. In Caco-2 cells, none of the compounds induced MET or EMT under our test conditions. In contrast, some products did induce MET in three other colon cancer cell lines (Figure 1). Among tested compounds, higher doses of fisetin and forskolin exhibited MET induction only in the KM12 cell line. Neither MET nor EMT could be observed in LoVo and WiDr cell lines with these compounds. On the other hand, treatment with 5 μM doses of apigenin and fulvestrant resulted in MET induction in LoVo but not in KM12 and WiDr (Appendix A). Therefore, even within the same cancer type, the compounds that can induce MET vary. We then evaluated EMT or MET in the breast cancer cell line MDA-MB-157, for which we previously showed phenotypic plasticity in 3D culture [59]. With the exception of resveratrol treatment (20 μM), most treatments resulted in the downregulation of CDH1, contrary to our expectations. Lastly, we treated a mesenchymal pancreatic cancer cell line, MIA PaCa-2, with various doses of natural compounds and repurposed agents. Almost all the treatments resulted in a two- to three-fold upregulation of CDH1 expression, possibly suggesting MET induction, however the VIM expression levels did not change. The increase in CDH1 expression may be considered as a mild phenotypic switch in the MIA PaCa-2 cells since CDH1 levels were already very low in this cell line when compared to breast and colon cell lines (Appendix A).

Overall, we noted transcriptional changes suggesting a shift towards a more epithelial phenotype in four out of the six cell lines tested upon treatment with at least one compound. The MET induction effects summarized in Appendix A show that none of the natural compounds or targeted agents could trigger MET in all cancer cells screened. In addition, compounds with clear or slight MET induction only caused these effects in a specific cell line, or only in a specific cancer type.

### 2.3. MET Induction by Natural Compounds Is Not Uniformly Related to Increased Sensitivity to Chemotherapeutics but Can Result in Occasional Synergistic or Additive Effects

We next asked whether the treatment with natural compounds would result in chemosensitization. For this purpose, the KM12 and WiDr colon cancer cell lines were treated with various doses of natural products, alone or in combination with two doses (1 μM and 10 μM) of 5-FU, a commonly used chemotherapeutic in colon cancer, after twenty-four hours of cell seeding. Cell viability assays were then performed 72 h after the treatments. In KM12, we observed MET induction with 20 uM of fisetin and two doses (10 μM and 30 μM) of forskolin. However, only the 10 uM 5-FU and forskolin combination treatments resulted in synergy, with the combination index (CI) score ranging from 0.91 to 0.37 (Appendix A). Additionally, 10 uM 5-FU and urolithin A combinations showed slight synergy, with the CI ranging from 0.95 to 0.77, despite no significant MET induction with urolithin A in this cell line. Interestingly, 5 μM fulvestrant and 5-FU combinations also resulted in synergy, with the CI ranging from 0.80 to 0.24. All the other combinations tested in KM12 showed additive effects or a slight antagonism. In the WiDr cell line, where we observed a mild induction of MET with forskolin only, the forskolin and 5-FU combinations showed slight antagonisms. Synergisms were observed with a combination of 1 μM 5-FU with various doses of apigenin, EGCG, and fisetin. In a similar way to the KM12 line, in the WiDr cell line, the fulvestrant and 5-FU combination resulted in a slight synergism. Additionally, we used the rectal cancer cell line SW837 in our analysis to examine the ability of natural products and repurposed agents to act as chemotherapy enhancers by combining them with irinotecan, which is a component of FOLFIRI routinely used in colorectal cancer, mainly as a second-line treatment [60,61]. Procyanidin B2, which did not have an effect on either the inhibition of cellular viability, or the induction of MET in the colon cancer cell lines, exhibited synergism in the SW837 line, with the CI ranging from 0.89 to 0.35. Additionally, specific doses of urolithin A, fulvestrant, and metformin, combined with irinotecan, also resulted in a slight synergism. In contrast, all other combinations showed either additive effects or even antagonism for SW837 (Appendix A).

We next investigated the combinatory effects of irinotecan, used in pancreatic cancer, and some widely studied natural compounds, curcumin and resveratrol, in pancreatic cancer cell lines, AsPC-1, BxPC-3, MIA PaCa-2, and SU8686 (Figure 2 and Figure 3). Most of the curcumin/irinotecan combinations resulted in antagonistic effects, while a 15 μM irinotecan and 20 μM curcumin combination showed synergy in AsPC-1 and SU8686 cell lines. Only the highest dose combinations of irinotecan and resveratrol resulted in a slight synergy in BxPC-3, MIA PaCa-2, and SU8686. The mild induction of MET by natural products in the MIA PaCa-2 cell line did not enhance irinotecan sensitivity.

Gemcitabine is a standard agent for the treatment of pancreatic cancer, either alone or in combination [62,63]. We therefore evaluated the combinatory effects of natural products with gemcitabine (Figure 4), as well as with the targeted agent gefitinib, an EGFR inhibitor, since EGFR inhibition has shown promising activity in pancreatic cancer, in the MIA PaCa-2 cell line (Figure 5) [64]. Six different combinations of gemcitabine and gefitinib were combined with a single dose of the natural products. Synergy was observed for only 10 μM of resveratrol, and 5 μM and 10 μM of gemcitabine combinations, with CI scores 0.71 and 0.47 (Figure 4), whereas no synergistic effect was noted for gefitinib (Appendix A). On the other hand, combinations of low dose gemcitabine and gefitinib, showed additive effects in MIA PaCa-2, however, the effect evolved from an additive to an antagonistic one at higher doses. The fact that synergy was observed with only gemcitabine among the three conventional drugs (irinotecan, gemcitabine, gefitinib) combined with resveratrol shows that the synergistic effects of a specific natural compound cannot be generalized even for one cell line; instead, the specific molecular cascades targeted by natural compound–drug pairs are most likely critical factors. Overall, these data indicate that there is no perfect overlap between MET-inducing and chemosensitizing compounds, which is likely to stem from the existence of more complex mechanisms underlying synergy patterns.

## 3. Discussion

Innate or acquired resistance of cancer cells to chemotherapeutics is responsible for tumor growth and metastasis [65,66]. Several studies have suggested that tumors exhibit a high degree of molecular and genetic heterogeneity, resulting in their adaptation to the conventional cytotoxic therapeutics. Alterations in the drug metabolic pathway, tumor evolution resulting in new mutations and thus the inhibition of cell death, epigenetic changes, mutations in the drug targets, and many other undiscovered mechanisms, cause resistance to cancer treatment. These mechanisms can act independently or in combination, and through various signaling pathways [67]. EMT is a well-known phenomenon whereby tumor cells lose their epithelial characteristics and acquire mesenchymal characteristics. Emerging evidence suggests a molecular and phenotypic association between EMT and chemotherapy resistance in several cancer types [68]. Therefore, targeting EMT has been considered a novel strategy to overcome therapy resistance in cancer.

A wide range of natural products have been reported as being cytotoxic for cancer cells and therefore useful in cancer therapy. Natural products can act as therapeutic, preventive, or chemosensitizer agents. Enhancement of chemotherapy sensitivity in tumor cells using naturally occurring biomolecules is a new and promising strategy aiming to promote the efficacy of conventional chemotherapeutics and targeted drugs [69]. Several studies suggest that inhibition or reversal of EMT by natural products results in resensitizing drug resistant cancer cells [15,21,22,27,36].

In the present study, we used eight natural compounds and two repurposed agents, that were previously shown to modulate EMT in different types of cancer cell lines, to test their capability to sensitize cancer cells to chemotherapy in a comprehensive manner. Although there are studies supporting the modulatory effects of all these compounds on EMT, only a few of them were able to induce MET in specific cancer cell lines in our experiments, suggesting a highly selective cell-line specific action. Interestingly, none of the compounds were able to induce MET in the Caco-2 cell line, which can easily be induced to undergo EMT and MET under various other conditions [58]. In fact, some treatments resulted in upregulation of VIM, suggesting the induction of EMT. Moreover, the MET induction with fisetin and forskolin seen in KM12 was not observed in other colon cancer cell lines. This suggests that MET induction with these natural compounds may vary according to cellular and molecular characteristics of cell lines even within the same cancer type. Similarly, the breast cancer cell line MDA-MB-157 exhibited MET only with resveratrol. Since various cellular mechanisms are effected by resveratrol, including mammalian target of rapamycin (mTOR) signaling, miRNA modulation, AMP-activated protein kinase (AMPK) activation, NF-B activation, nitric oxide production, histone deacetylase (HDAC) inhibition, and cytochrome P450 inhibition [70], it might be that MET induced in this cell line is managed through one or more of these mechanisms which may not have been targeted or effected sufficiently by the other compounds in our panel.

Different transcriptomic signatures have been evaluated for determining the EMT status of cancer cells [71,72,73]. Evaluation of the expression of both CHD1 and VIM genes was shown to be a highly reliable and consistent predictor of the EMT phenotype [72,74,75,76]. Our experience also shows that CDH1/VIM-based evaluation of EMT is very reliable [59]. Throughout this study, induction of MET has been assessed based on a reduction in expression of VIM, and an increase in the expression of CDH1 upon treatment. Exceptionally, MIA PaCa-2, which had a very low basal expression of CDH1, had a very dramatic increase in CDH1 expression induced by multiple natural compounds. Almost no change was recorded for VIM expression, therefore we did not consider these changes as markers of MET in this cell line. Evaluation of changes in the expression of multiple mesenchymal markers may be needed to better identify if these compounds can induce MET in this cell line.

Among the compounds we examined in this study, almost none were found to both trigger MET and induce chemosensitization. In KM12, although fisetin and forskolin induced MET, the fisetin and 5-FU combination did not show a synergistic effect. On the other hand, urolithin A synergized with 5-FU in KM12 in the absence of MET induction. Furthermore, slight MET induction in MIA PaCa-2 with natural compounds did not lead to improved sensitivity to irinotecan, gemcitabine, and gefitinib in almost all combinations.

These findings show that synergy between natural compounds and chemotherapeutics/targeted agents can be observed only for some cell lines and that the underlying mechanism is likely not to be the simple induction of MET, but possibly a combination of several other mechanisms. The fact that synergistic effects were also quite variable across cancer types as well as in cell lines within a given cancer type suggests that the mechanisms underlying these observations are highly cell line specific.

## 4. Materials and Methods

### 4.1. Cell Lines and Culture Conditions

Breast cancer cell lines, CAL51, MDA-MB-157, MDA-MB-231, MDA-MB-436, MDA-MB-453, and MDA-MB-468 were cultured in DMEM medium (Biowest, MO, USA). Colon cancer cell lines, Caco-2, HCT-8, KM-12, LS513, LoVo, SW837, and WiDr were cultured in RPMI-1640 medium (Biowest, MO, USA). Pancreatic cancer cell lines, AsPC-1, BxPC-3, and SU8686 were cultured in RPMI-1640 medium, while MIA PaCa-2 cells were cultured in DMEM medium. The respective media were supplemented with 10% fetal bovine serum (FBS) (Biowest, MO, USA), 1% penicillin-streptomycin (Lonza, Switzerland), and 1% 200 mM L-Glutamine (Lonza, Switzerland) at 37 °C with 5% CO_2_.

### 4.2. Chemicals

Chemotherapeutics, 5-FU (S1224), gemcitabine (S1149) and irinotecan (S2217), were purchased from Selleckchem (Houston, TX, USA). Targeted cancer therapy drugs, gefitinib (S1025) and selumetinib (S1008) were purchased from Selleckchem, while lapatinib (11493) was purchased from Cayman Chemical (Ann Harbor, MI, USA). Fulvestrant (S1191, Selleckchem) and metformin (13118, Cayman Chemical) were used as repurposed agents. Natural products, apigenin (S2262), curcumin (S1848), fisetin (S2298), and forskolin (S2449) were purchased from Selleckchem, while (-)- epigallocatechin gallate (70935), procyanidin B2 (19865), resveratrol (70675) and urolithin A (22607) were purchased from Cayman Chemical. All chemicals had more than 98% purity. Stock solutions were prepared according to the manufacturers’ instructions.

### 4.3. Cell Viability Assay

Cancer cell lines were seeded at 2000 cells per well in 96-well plates (Corning, NY, USA). After incubation for 24 h, cells were treated with chemotherapeutics, targeted drugs, and natural compounds using 6 different concentrations: apigenin, curcumin, EGCG, fisetin, forskolin, procyanidin B2, resveratrol, urolithin A, fulvestrant: 0.1–100 μM; metformin: 0.4–60 mM (in colon and pancreatic cancers); metformin: 0.1–100 μM (in breast cancer). Cell viability was analyzed in quadruplicates using the CellTiter-Glo Luminescent Cell Viability Assay (Promega, WI, USA), according to the manufacturer’s protocol, after 72 h of treatment. IC50 values were obtained by a method previously described [59]. Different methods were used to assess combination treatments. The first approach was the combination of a single dose of a natural compound (resveratrol: 10 μM; procyanidin B2: 20 μM; forskolin: 40 μM; fisetin: 20 μM; fulvestrant: 10 μM; metformin: 2.5 mM) with six different doses of a chemotherapeutic (gemcitabine: 1–100 μM) or targeted agent (gefitinib: 0.1–100 μM) in the MIA PaCa-2 cell line. In this approach, a total of 2000 cells were seeded in quadruplicates. In the second approach, four different doses of natural compounds were tested with two different doses of chemotherapeutics (5-FU: 1 μM and 10 μM; irinotecan: 2 μM and 40 μM). In this approach, a total of 2000 cells were seeded in duplicates for KM12 and WiDr cell lines and in quadruplicates for the SW837 cell line. In the third approach, six different doses of natural compounds (curcumin and resveratrol: 2.5–80 μM) were combined with six different doses of irinotecan: 0.186–60 μM in pancreatic cancer cell lines, AsPC-1, BxPC-3, MIA PaCa-2, and SU8686. In this approach, a total of 2000 cells were seeded in triplicates. All combination treatments were administered to cancer cell lines in 96-well plates after the cell lines were seeded and incubated for 24 h. After 72 h treatment, cell viability was quantified with the CellTiter-Glo Luminescent Cell Viability Assay.

### 4.4. Combination Index Analysis

To quantitate synergy combination treatments of natural compounds, chemotherapeutics and targeted drugs were analyzed with the Chou-Talalay methods using CompuSyn Software (Paramus, NJ, USA) which generates a combination index (CI) which was calculated with the cell viability values from combination treatments. CI < 1 indicates synergy, CI = 1 indicates additive effect, while CI > 1 indicates antagonism.

### 4.5. Quantitative Real Time PCR Analysis 

Total RNA was isolated using Purezol (Biorad, CA, USA) according to the manufacturer’s instructions. Two micrograms of total RNA were reverse-transcribed into cDNA using the Revert-aid first strand cDNA synthesis kit (Thermo Fisher Scientific, MA, USA) according to the supplier’s protocol using a random hexamer. Quantitative profiling of EMT marker genes CDH1 and VIM by qRT-PCR was performed in triplicates using a LightCycler 480 II real-time PCR system (Roche, Switzerland) and iTaq Universal SYBR Green Supermix (Bio-rad, CA, USA). Real-time PCR reactions were run under cycling conditions according to the manufacturer’s instructions. GAPDH was used as an endogenous control in qPCR reactions in triplicates and gene expression signals were normalized to GAPDH. Relative expression values of CDH1 and VIM were calculated by the ddCt method according to samples with least expression as previously described [77]. qPCR primer pairs used in this study were as follows: GAPDH_F: GCTCATTTCCTGGTATGACAACG, GAPDH_R: CTCTTGTGCTCTTGCTGGGG, CDH1_F: TCCAGGAACCTCTGTGATGGA, CDH1_R: CGTAGGGAAACTCTCTCGGTC, VIM_F: CGGGAGAAATTGCAGGAGGA, and VIM_R: AAGGTCAAGACGTGCCAGAG.

### 4.6. Statistical Analysis

Statistical analysis was performed using R Statistical Software (R Foundation, Vienna, Austria) and different treatment groups were compared using the Mann–Whitney U test. *p*-values less than 0.05 were considered as statistically significant.

## Figures and Tables

**Figure 1 molecules-26-05858-f001:**
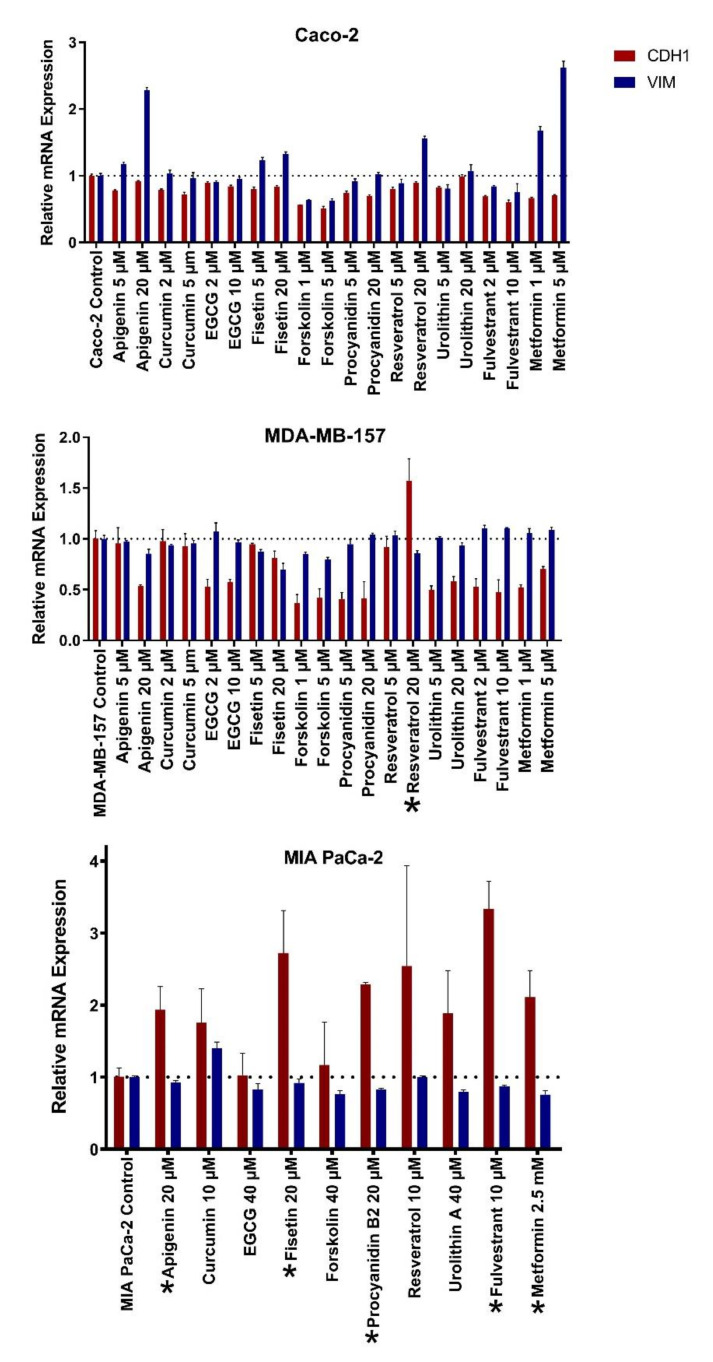
Effect of natural compounds and repurposed agents on the expression of EMT marker genes CDH1 and VIM. Cells were treated with indicated concentrations for 48 h. Gene expression levels were then evaluated by qRT-PCR and normalized to GAPDH levels. Data are presented as the mean with standard deviation for three replicates. Asterisks (*) before compound names indicate significant upregulation in CDH1 or downregulation in VIM expressions according to the Mann–Whitney U test; * indicates *p*-value < 0.05.

**Figure 2 molecules-26-05858-f002:**
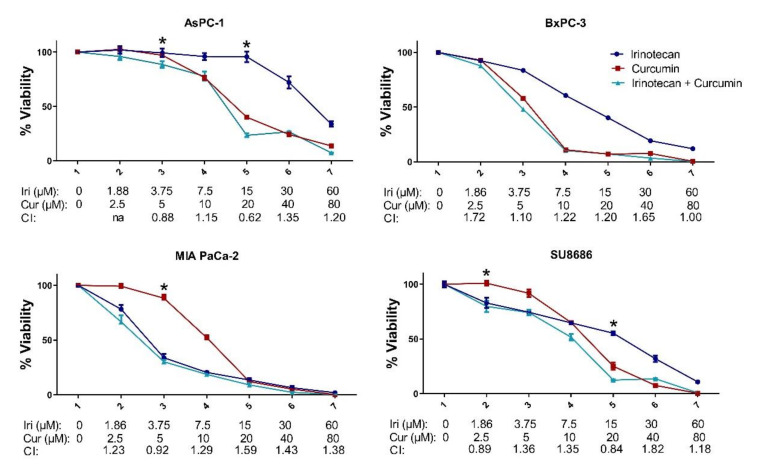
Combination treatment of curcumin and irinotecan in pancreatic cancer cell lines. The cells were treated with indicated concentrations for 72 h. Data are presented as the mean with standard deviation of 4 replicates. CI = combination index value; * shows combinations with CI < 1 indicating synergy. Other combinations resulted in additive or antagonistic effects.

**Figure 3 molecules-26-05858-f003:**
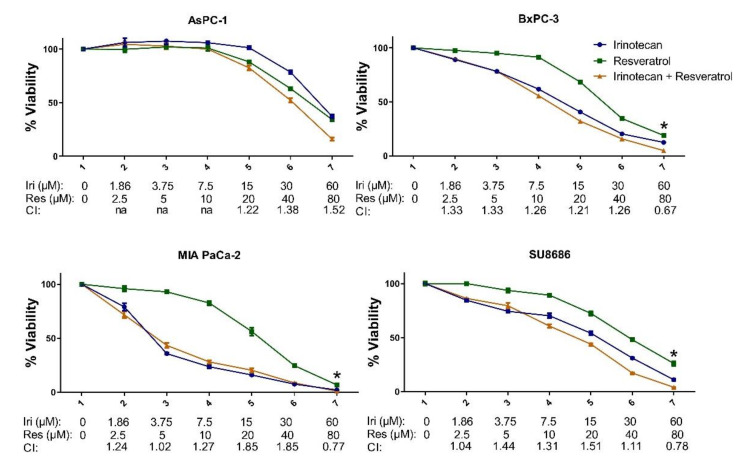
Combination treatment of resveratrol and irinotecan in pancreatic cancer cell lines. The cells were treated with indicated concentrations for 72 h. Data are presented as the mean with standard deviation of 4 replicates. CI = combination index value; * shows combinations with CI < 1 indicating synergy. Other combinations resulted in additive or antagonistic effects.

**Figure 4 molecules-26-05858-f004:**
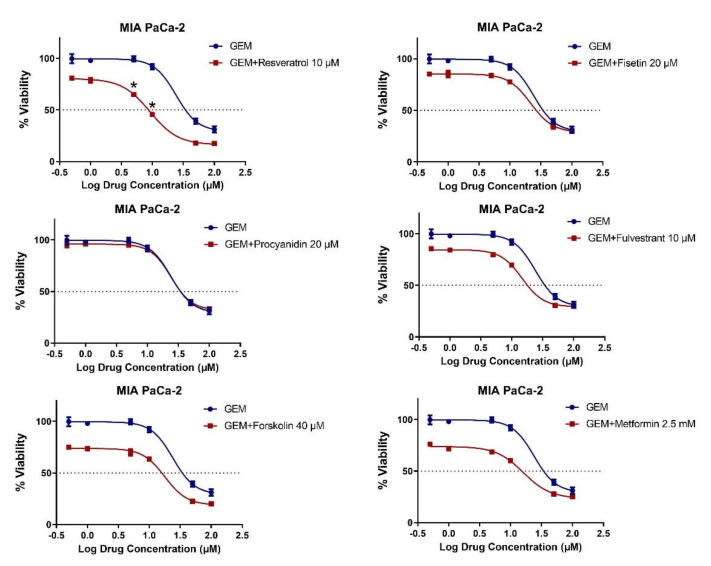
Combination treatment of natural compounds and gemcitabine in the MIA PaCa-2 pancreatic cancer cell line. The cells were treated with indicated concentrations for 72 h. Data were presented as the mean with standard deviation of 4 replicates. * shows combinations with CI < 1 indicating synergy. Other combinations resulted in additive (CI = 1) or antagonistic effects (CI > 1).

**Figure 5 molecules-26-05858-f005:**
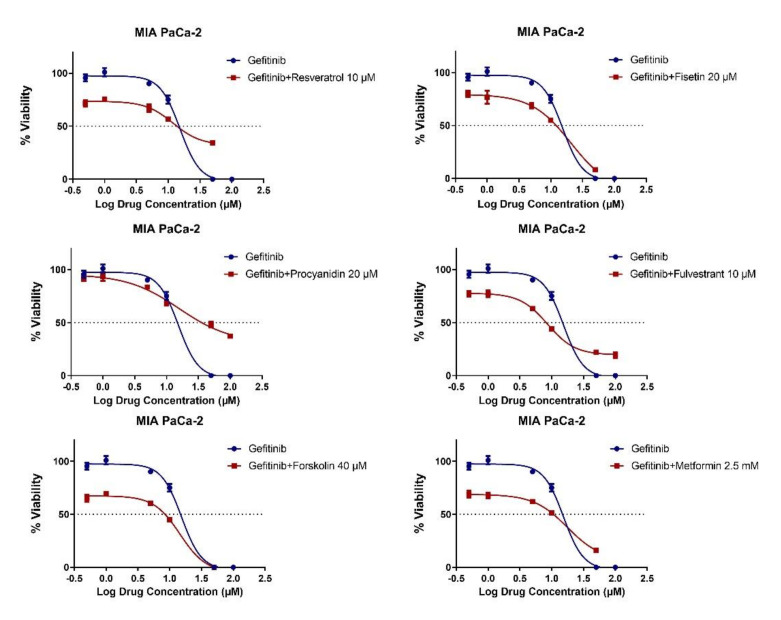
Combination treatment of natural compounds and gefitinib in the MIA PaCa-2 pancreatic cancer cell line. The cells were treated with indicated concentrations for 72 h. Data are presented as the mean with standard deviation of 4 replicates. All combinations resulted in additive (CI = 1) or antagonistic effects (CI > 1).

**Table 1 molecules-26-05858-t001:** In vitro cytotoxicity analyses of natural compounds and repurposed agents in breast cancer cell lines.

	CAL-51	MDA-MB-157	MDA-MB-231	MDA-MB-436	MDA-MB-453	MDA-MB-468
Apigenin	34 ± 2.9 *	78 ± 4.6	>100	52 ± 4.2	36 ± 5.8	43 ± 13
Curcumin	24 ± 3.4	23 ± 1.2	40 ± 2.7	12 ± 2.2	14 ± 3.2	30 ± 3.8
EGCG	80 ± 3.2	>100	79 ± 3.6	>100	85 ± 3.8	83 ± 3.1
Fisetin	41 ± 7.6	39 ± 2.1	>100	86 ± 9.3	72 ± 5.1	>100
Forskolin	48 ± 3.3	>100	>100	>100	38 ± 5.3	>100
Procyanidin B2	>100	>100	>100	>100	>100	>100
Resveratrol	59 ± 4.4	65 ± 5.6	33 ± 2.6	54 ± 5.7	64 ± 7.8	30 ± 5.7
Urolithin A	>100	>100	>100	>100	57 ± 3.3	>100
Fulvestrant	>100	>100	>100	>100	29 ± 7.5	>100
Metformin(μM)	>100	>100	>100	>100	>100	>100

* Values correspond to IC50 ± standard error of mean (μM).

**Table 2 molecules-26-05858-t002:** In vitro cytotoxicity analyses of natural compounds and repurposed agents in colon cancer cell lines.

	HCT8	KM12	LS513	LoVo	SW837	WiDr
Apigenin	19 ± 2.9 *	26 ± 2.3	12 ± 3.9	28 ± 6.8	50 ± 3.6	40 ± 8
Curcumin	12 ± 1.1	10 ± 1.9	6 ± 0.6	9 ± 0.6	15 ± 1.6	8 ± 0.5
EGCG	14 ± 2.6	17 ± 1.7	17 ± 1.9	23 ± 2.9	15 ± 2.1	20 ± 2.9
Fisetin	35 ± 6.9	16 ± 3.1	26 ± 3.3	24 ± 3.1	38 ± 4.3	66 ± 7.9
Forskolin	>100	4 ± 0.3	>100	>100	>100	>100
Procyanidin B2	>100	>100	>100	>100	>100	>100
Resveratrol	35 ± 5.1	47 ± 4.6	24 ± 1.9	27 ± 3.3	25 ± 1.8	39 ± 2.7
Urolithin A	40 ± 3.3	>100	30 ± 3.1	28 ± 2.7	40 ± 2.2	52 ± 3
Fulvestrant	>100	>100	>100	>100	>100	>100
Metformin(mM)	2 ± 1.2	13 ± 4.1	2 ± 2.1	10 ± 2.5	23 ± 2.7	10 ± 3.3

* Values correspond to IC50 ± standard error of mean (μM).

**Table 3 molecules-26-05858-t003:** In vitro cytotoxicity analyses of natural compounds and repurposed agents in pancreatic cancer cell lines.

IC50 (μM)	AsPC-1	BxPC-3	MIA PaCa-2	SU8686
Apigenin	34 ± 1.3 *	18 ± 2.3	40 ± 5.4	21 ± 0.9
Curcumin	18 ± 2.2	7 ± 0.4	11 ± 0.8	17 ± 1.7
EGCG	35 ± 7.6	26 ± 3.7	87 ± 10.2	22 ± 2.4
Fisetin	90 ± 7.7	25 ± 1.1	38 ± 4.7	45 ± 11.9
Forskolin	>100	45 ± 3.1	>100	>100
Procyanidin B2	>100	>100	>100	>100
Resveratrol	65 ± 4.4	32 ± 1.7	23 ± 1.8	40 ± 4.1
Urolithin A	>100	92 ± 4.8	96 ± 1.8	>100
Fulvestrant	>100	>100	>100	>100
Metformin(mM)	40 ± 2.1	7 ± 1.2	27 ± 3.7	4 ± 5.5

* Values correspond to IC50 ± standard error of mean (μM).

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
