# Peer review of "Epithelial-to-Mesenchymal Transition Is Not a Major Modulating Factor in the Cytotoxic Response to Natural Products in Cancer Cell Lines"

_molecules, 2021, doi:10.3390/molecules26195858_

Round 1

Reviewer 1 Report

28 July 2021

Review 2

Manuscript: Epithelial-to-mesenchymal transition is not a major modulating factor in the cytotoxic response to natural products in cancer cell lines

Authors: B Kucukkaraduman, EG Cicek, MW Akbar, SD Canli, B Vural, AO Gure

Journal: Molecules

General comment:

I read the new version of the manuscript and the author’s reply to the first review. Some of the first review comments have not been appropriately resolved and, after read the author’s reply, I have the security that the authors have not intention to make it.

Specific comments:

1. First review comment 1: “.. in my opinion, is needed that the authors demonstrated that the effects described on cell viability are selective or specific on cancer cell lines. What are the effects on non-cancer cell lines? If they are the same, the results have not validity for the last aim of the work.”

Authors reply: “The aim of this study was to test in a comprehensive manner a hypothesis which is generally accepted as a fact by the scientific community; that is that natural compounds can induce EMT in cancer cell lines and that this correlates with increased sensitivity to therapy. The significance of our findings is that this hypothesis is not correct at a general scale. And that if there is a relation between these two concepts it needs to be studied in much more detail. We have shown how such a study can be performed in another study of ours which has recently been published (Akbar et al. 2020, Journal of Cancer 11, 949).”

Second review comment. This comment has not been appropriately addressed by the authors. The authors do not demonstrated that the effects of different natural compounds used in these experimental conditions are selective or specific on cancer cell lines.

2. First review comment 2: “Pages 3 and 4, Tables 1-3. The authors described the effects of different compounds on cell viability of several cancer human cell lines (breast, colon, pancreatic). For having sure that the effects of each compound is specific or selective on the different cancerous cells, is needed to compare with the effect of each compound on a non-cancerous cell line (breast tissue, colon or intestinal cell; pancreatic cells). So, in each table, one or two new columns must be added in which the effects of each compound will be described. Only the compounds that have differential effects on the viability of both types of cell lines must be considered.”

Authors reply: “If the hypothesis that natural compound induced MET increases chemosensitivity had been validated here, we would have suggested the readers to use low doses of natural compounds (not toxic for any cell) and combine this with therapy. Since the hypothesis has not been validated, we have not planned additional experiments. There are several studies showing that natural compounds do not cause any damage to normal cells in low levels. (PMID: 15367704, PMID: 29034071, PMID: 31261976)

Second review comment. This aspect has not been appropriately addressed by the authors. The authors do not demonstrated that the effects of different natural compounds used were selective or specific on cancer cell lines. The indicated references are related only with three natural compounds (curcumin, apigenin, fisentin) on specific type of cells (hepatocytes, other, hepatocellular carcinoma cells, respectively). Nothing to see with the experiment described here.

3. First review comment 3: “In my opinion, in the actual form, it is very difficult to follow and knowledge the results reported. Many compounds have been tested in many cell lines but so it is very difficult to extract some concrete conclusion. Authors must clarify what are the most important findings showed by this manuscript about the selective effects of the differents compounds tested.”

Authors reply: “Thank you for this comment. We have now included several summary tables into our manuscript.”

Second review comment. I had look for into the manuscript for the new summary tables and I had not found there. Tables and Supplementary Tables are the same that in the first version of the manuscript.

4. First review comment 5: “Page 1, abstract, lines 15 and 17, sentences: “Natural products exhibit…” and “… effects of natural compounds…” These sentences are very imprecise and ambiguous. These sentences must be rewritte to concrete and avoid imprecise description.”

Authors reply: “As the abstract is only a general description of the study we have not changed the sentences.”

Second review comment: This aspect has not been appropriately addressed by the authors. What natural products exhibit antiproliferative activity against cancer cells? All of them? What natural compounds have been investigated? These sentences need to be clarify in the Abstract -the more read section of the manuscript- for avoid imprecise description.

5. First review comment: “Page 4, lines 157-158. “An increase in CDH1, …and viceversa for EMT”. This affirmation must be justified appropriately”

Authors reply: “The sentence explains the choice of CHD1 and VIM expression levels as indicators of EMT and MET.”

Second review comment: This aspect has not been appropriately addressed by the authors. This affirmation must be appropriately explained and justified. The explanation and justification must be added to the manuscript because is central in the subsequent interpretations. In the actual version it is not justified.

6. First review comment: “8. Statistical treatment of the results must be added into the tables and figures. Results must be expressed as mean ± standard error of the mean (for example) and the statistical significance for the different comparisons must be added.”

Authors reply: “The figures where Mann-Whitney U test were performed are now indicated.”

Second review comment: Authors change the statistical test applied but figures and tables continues practically as in the first version of the manuscript. Statistical treatment of the results has not been added to tables and figures.

7. First review comment: “9. The sections Results and Discussion must be unified as a only one “Results and Discussion” section. In the actual Results section it has been included explicative paragraph (for example: page 4, first paragraph of 2.2 subsection) no proper of this section.”

Authors reply: “The structure of the manuscript needs to follow the publisher’s guidelines. This is what we have done.”

Second review comment: This aspect has not been appropriately addressed by the authors.

8. First review comment: “10. Page 12, Subsections 4.3-4.5. In all these subsections the number of independent repetitions (replicates) of the experiments for each treatment must be clearly indicated.”

Authors reply: “These have been clarified.”

Second review comment: I read 4.3-4.5 subsections and I had not found text added for clarification. The text are practically the same as in the first version of the manuscript.

Author Response

Reviever 1:

28 July 2021

Manuscript: Epithelial-to-mesenchymal transition is not a major modulating factor in the cytotoxic response to natural products in cancer cell lines

Authors: B Kucukkaraduman, EG Cicek, MW Akbar, SD Canli, B Vural, AO Gure

Journal: Molecules

General comment:

I read the new version of the manuscript and the author’s reply to the first review. Some of the first review comments have not been appropriately resolved and, after read the author’s reply, I have the security that the authors have not intention to make it.

Specific comments:

  1. First review comment 1: “.. in my opinion, is needed that the authors demonstrated that the effects described on cell viability are selective or specific on cancer cell lines. What are the effects on non-cancer cell lines? If they are the same, the results have not validity for the last aim of the work.”

Authors reply: “The aim of this study was to test in a comprehensive manner a hypothesis which is generally accepted as a fact by the scientific community; that is that natural compounds can induce EMT in cancer cell lines and that this correlates with increased sensitivity to therapy. The significance of our findings is that this hypothesis is not correct at a general scale. And that if there is a relation between these two concepts it needs to be studied in much more detail. We have shown how such a study can be performed in another study of ours which has recently been published (Akbar et al. 2020, Journal of Cancer 11, 949).”

Second review comment. This comment has not been appropriately addressed by the authors. The authors do not demonstrate that the effects of different natural compounds used in these experimental conditions are selective or specific on cancer cell lines.

Answer: We still do not understand why this question is relevant.  What does the reviewer mean when he/she uses the terms “selective” and “specific”? If the question is: do these compounds affect normal (non-cancer) cells? then our response is: this question is irrelevant to this work as there are a number of papers where these compounds have been tested on cancer cells, as well as on normal cells (e.g. Breast-Fisetin-PMID: 26755433; Colon-EGCG- PMID: 29596305). The question we have is not whether they are also effective on normal cells. Our questions are (1) “do they cause MET in cancer; as has been argued in the literature”, and  (2)  “if they do cause MET, does this correlate with an increased sensitivity to chemotherapeutics; as has been argued in the literature”.

  1. First review comment 2: “Pages 3 and 4, Tables 1-3. The authors described the effects of different compounds on cell viability of several cancer human cell lines (breast, colon, pancreatic). For having sure that the effects of each compound is specific or selective on the different cancerous cells, is needed to compare with the effect of each compound on a non-cancerous cell line (breast tissue, colon or intestinal cell; pancreatic cells). So, in each table, one or two new columns must be added in which the effects of each compound will be described. Only the compounds that have differential effects on the viability of both types of cell lines must be considered.”

Authors reply: “If the hypothesis that natural compound induced MET increases chemosensitivity had been validated here, we would have suggested the readers to use low doses of natural compounds (not toxic for any cell) and combine this with therapy.  Since the hypothesis has not been validated, we have not planned additional experiments. There are several studies showing that natural compounds do not cause any damage to normal cells in low levels. (PMID: 15367704, PMID: 29034071, PMID:  31261976)

Second review comment. This aspect has not been appropriately addressed by the authors. The authors do not demonstrate that the effects of different natural compounds used were selective or specific on cancer cell lines. The indicated references are related only with three natural compounds (curcumin, apigenin, fisetin) on specific type of cells (hepatocytes, other, hepatocellular carcinoma cells, respectively). Nothing to see with the experiment described here.

Answer: Please see our response to the first question.

  1. First review comment 3: “In my opinion, in the actual form, it is very difficult to follow and knowledge the results reported. Many compounds have been tested in many cell lines but so it is very difficult to extract some concrete conclusion. Authors must clarify what are the most important findings showed by this manuscript about the selective effects of the differents compounds tested.”

Authors reply: “Thank you for this comment. We have now included several summary tables into our manuscript.”

Second review comment. I had look for into the manuscript for the new summary tables and I had not found there. Tables and Supplementary Tables are the same that in the first version of the manuscript.

Answer: Important findings are summarized in lines 126-132 and lines 166-171. Tables S1 and S3 are the summary tables and show all results.

  1. First review comment 5: “Page 1, abstract, lines 15 and 17, sentences: “Natural products exhibit…” and “… effects of natural compounds…” These sentences are very imprecise and ambiguous. These sentences must be rewritten to concrete and avoid imprecise description.”

Authors reply: “As the abstract is only a general description of the study, we have not changed the sentences.”

Second review comment: This aspect has not been appropriately addressed by the authors. What natural products exhibit antiproliferative activity against cancer cells? All of them? What natural compounds have been investigated? These sentences need to be clarified in the Abstract -the more read section of the manuscript- for avoid imprecise description.

Answer:  All compounds are anti-proliferative for at least one cell line, at least at one dose tested. We modified the abstract to include names of compounds tested and summary statements of our findings. 

  1. First review comment: “Page 4, lines 157-158. “An increase in CDH1, …and viceversa for EMT”. This affirmation must be justified appropriately”

Authors reply: “The sentence explains the choice of CHD1 and VIM expression levels as indicators of EMT and MET.”

Second review comment: This aspect has not been appropriately addressed by the authors. This affirmation must be appropriately explained and justified. The explanation and justification must be added to the manuscript because is central in the subsequent interpretations. In the actual version it is not justified.

Answer: Transcriptomic based EMT phenotype scoring is reliable (Tan et al. 2014, EMBO Mol Med). Many studies have used CDH1 and VIM, two genes highly expressed in epithelial and mesenchymal cell states respectively, to characterize EMT status (Park et al. 2008, Genes&Dev, Qiu et al. 2020, RNA, Chakraborty et al. 2020, Front. Bioeng. Biotechnol). Evaluation of the CDH1/VIM gene pair has been shown to be a most reliable predictor of EMT for cell lines (George et al. 2017, Cancer Res).   Our experience also shows that CDH1/VIM based evaluation of EMT is very reliable (Akbar et al. 2020, Journal of Cancer 11, 949).

We have added a paragraph to the discussion section summarizing these observations, and therefore why these two genes were selected in our work for the evaluation of EMT.

  1. First review comment: “8. Statistical treatment of the results must be added into the tables and figures. Results must be expressed as mean ± standard error of the mean (for example) and the statistical significance for the different comparisons must be added.”

Authors reply: “The figures where Mann-Whitney U test were performed are now indicated.”

Second review comment: Authors change the statistical test applied but figures and tables continue practically as in the first version of the manuscript. Statistical treatment of the results has not been added to tables and figures.

Answer: We have not “changed” the statistical test. We have only added the name of the test to the figures, as that is what we understood the reviewer requested. Statistical test were applied to Figure 1 and Figure S2 where asterisks indicate significant results. In Figure 2, Figure 3, Figure 4, Figure 5 and Table S2, the statistical test is the combination index (CI) scoring in which CI scores are indicated.

  1. First review comment: “9. The sections Results and Discussion must be unified as a only one “Results and Discussion” section. In the actual Results section, it has been included explicative paragraph (for example: page 4, first paragraph of 2.2 subsection) no proper of this section.”

Authors reply: “The structure of the manuscript needs to follow the publisher’s guidelines. This is what we have done.”

Second review comment: This aspect has not been appropriately addressed by the authors.

Answer: We removed the introductory statement at the beginning of section 2.2.

  1. First review comment: “10. Page 12, Subsections 4.3-4.5. In all these subsections the number of independent repetitions (replicates) of the experiments for each treatment must be clearly indicated.”

Authors reply: “These have been clarified.”

Second review comment: I read 4.3-4.5 subsections and I had not found text added for clarification. The text are practically the same as in the first version of the manuscript.

Answer: This information has now been added to section 4.3 and 4.5.

Reviewer 2 Report

Authors examined IC50 value of each product. But, I wonder if the products increase the cell death or decrease the cell growth. If cell death is induced, authors must clarify what types of cell death occur (apoptosis, necroptosis or others)?

Author Response

Reviewer 2:

Authors examined IC50 value of each product. But, I wonder if the products increase the cell death or decrease the cell growth. If cell death is induced, authors must clarify what types of cell death occur (apoptosis, necroptosis or others)?

Answer: If the hypothesis that natural compound induced MET increases chemosensitivity had been validated here, we would have suggested the readers to use low doses of natural compounds (not toxic for any cell) and combine this with therapy. So, cell viability analyses were performed to identify lowest doses to induce MET. Our aim is not to identify toxicity profiles of natural compounds on cancer cell lines.

Reviewer 3 Report

The title is too big to carry natural products too far. The authors just used 10 drugs, how can be to represent all of the natural products.
The authors also didn’t explain that why do they chose these drugs for testing, what is the principle, what are other publications describe the anti-tumor effects on drugs?
For tables 1,2 and 3, the authors should include a relative normal cell line as a safety control.
The manuscript only used QPCR to examine two gene expressions to claim the EMT progress, it is too weak. No examination on key proteins of EMT, no functional assay, such as cell migration, cell invasion, immunostaining of key genes.
For figures 2-5, the authors didn’t have a single drug treatment, which is necessary.
Basically, the authors only tested drugs on cell viability of serval tumor cell lines, it is complete dissociation of EMT, even for drugs effect on tumor growth, no molecular mechanism explained the results of anti-tumor growth. 

Author Response

Reviewer 3:

The title is too big to carry natural products too far. The authors just used 10 drugs, how can be to represent all of the natural products.

Answer: We agree with the comment. However, we prefer not to list the names of all compounds in the title as this would make it a very long one, and now they are listed in the abstract.

The authors also didn’t explain that why do they chose these drugs for testing, what is the principle, what are other publications describe the anti-tumor effects on drugs?

Answer: In lines 44-78, we explain why we selected these natural compounds and how they affect EMT phenotype in cancer cell lines and act as a sensitizer to chemotherapeutics with references.  

For tables 1,2 and 3, the authors should include a relative normal cell line as a safety control.

Answer: If the hypothesis that natural compound induced MET increases chemosensitivity had been validated here, we would have suggested the readers to use low doses of natural compounds (not toxic for any cell) and combine this with therapy. Since the hypothesis has not been validated, we have not planned additional experiments. There are several studies showing that natural compounds do not cause any damage to normal cells in low levels. (PMID: 15367704, PMID: 29034071, PMID: 31261976).

The manuscript only used QPCR to examine two gene expressions to claim the EMT progress, it is too weak. No examination on key proteins of EMT, no functional assay, such as cell migration, cell invasion, immunostaining of key genes.

Answer: Transcriptomic based EMT phenotype scoring is reliable (Tan et al. 2014, EMBO Mol Med). Many studies have used CDH1 and VIM, two genes highly expressed in epithelial and mesenchymal cell states respectively, to characterize EMT status (Park et al. 2008, Genes&Dev, Qiu et al. 2020, RNA, Chakraborty et al. 2020, Front. Bioeng. Biotechnol). Evaluation of the CDH1/VIM gene pair has been shown to be a most reliable predictor of EMT for cell lines (George et al. 2017, Cancer Res).   Our experience also shows that CDH1/VIM based evaluation of EMT is very reliable (Akbar et al. 2020, Journal of Cancer 11, 949).

We have added a paragraph to the discussion section summarizing these observations, and therefore why these two genes were selected in our work for the evaluation of EMT.

For figures 2-5, the authors didn’t have a single drug treatment, which is necessary.

Answer: As these experiments are evaluating synergy which we calculate as the CI value, a single drug treatment is not necessary. However, all drugs were tested individually as given in Tables 1, 2 and 3.

Basically, the authors only tested drugs on cell viability of serval tumor cell lines, it is complete dissociation of EMT, even for drugs effect on tumor growth, no molecular mechanism explained the results of anti-tumor growth.

Answer: Our aim was not a discussion of the molecular mechanisms that cause cytotoxicity. We asked whether natural compounds could synergize with given treatment regimens.  If we had observed synergy, we might have added experiments to determine the mechanisms involved in this observation.

Reviewer 4 Report

Key remarks

The authors use a large amount of factual material, not always justifying the need for its applicationIC50 must be subscript.

Tables must be formatted so that the line with names of cell lines are in one line

Fulvestrant and metformin are not natural compounds. What is the need to use these compounds in an article about natural compounds? Moreover, these compounds do not possess cytotoxic activity on the studied lines?

Activities should be of the same level. If it is indicated that the activity of substances is higher than 100 micromoles, why for which compound give activity in millimoles?

The main conclusion of the article – induction of EMT or MET by combination of several other mechanisms, then discussed in the article. And then what is the concrete practical conclusion from the article?

Authors do not use enough modern literature on the topic, for example: Jonckheere S, Adams J, De Groote D, Campbell K, Berx G, Goossens S. Epithelial-Mesenchymal Transition (EMT) as a Therapeutic Target. Cells Tissues Organs. 2021 Jan 5:1-26. doi: 10.1159/000512218, Cheng F, Dou J, Zhang Y, Wang X, Wei H, Zhang Z, Cao Y, Wu Z. Urolithin A Inhibits Epithelial-Mesenchymal Transition in Lung Cancer Cells via P53-Mdm2-Snail Pathway. Onco Targets Ther. 2021 May 17;14:3199-3208. doi: 10.2147/OTT.S305595

With a lot of practical material, the article leaves more questions than answers. May be published only after serious major revision.

Author Response

Reviewer 4:

The authors use a large amount of factual material, not always justifying the need for its application

Answer: All natural compounds and drugs we tested were selected based on previous publications.

IC50 must be subscript.

Answer: We find the use of both formats in the literature.

Tables must be formatted so that the line with names of cell lines are in one line

Answer: We have formatted Tables 2 and 3 as such. But cell line names in Table 1 are too long and therefore need to be formatted as two lines. We do not see this causing a confusion.

Fulvestrant and metformin are not natural compounds. What is the need to use these compounds in an article about natural compounds? Moreover, these compounds do not possess cytotoxic activity on the studied lines?

Answer: These compounds can be repurposed as we indicated in the paper. Fulvestrant is cytotoxic for a breast cancer cell line (MDA-MB-453) (Table 1), while Metformin is cytotoxic for all colon and pancreas cancer cell lines (Tables 2 and 3).

Activities should be of the same level. If it is indicated that the activity of substances is higher than 100 micromoles, why for which compound give activity in millimoles?

Answer: Only metformin was screened at millimolar levels since its cytotoxicity was shown to be at these concentrations in literature. All other values are in micromolar format.  

The main conclusion of the article – induction of EMT or MET by combination of several other mechanisms, then discussed in the article. And then what is the concrete practical conclusion from the article?

Answer: The conclusion of the paper is that epithelial-to-mesenchymal transition is not a major modulating factor in the cytotoxic response to natural products in cancer cell lines.

Authors do not use enough modern literature on the topic, for example: Jonckheere S, Adams J, De Groote D, Campbell K, Berx G, Goossens S. Epithelial-Mesenchymal Transition (EMT) as a Therapeutic Target. Cells Tissues Organs. 2021 Jan 5:1-26. doi: 10.1159/000512218, Cheng F, Dou J, Zhang Y, Wang X, Wei H, Zhang Z, Cao Y, Wu Z. Urolithin A Inhibits Epithelial-Mesenchymal Transition in Lung Cancer Cells via P53-Mdm2-Snail Pathway. Onco Targets Ther. 2021 May 17;14:3199-3208. doi: 10.2147/OTT.S305595

Answer: We have now added these articles to the paper where they are relevant.

Round 2

Reviewer 1 Report

08 September 2021

Review 3

Manuscript: Epithelial-to-mesenchymal transition is not a major modulating factor in the cytotoxic response to natural products in cancer cell lines

Authors: B Kucukkaraduman, EG Cicek, MW Akbar, SD Canli, B Vural, AO Gure

Journal: Molecules

General comment:

Review 3: The majority of the comments have been appropriately answered. 

Specific comments:

  1. First review comment 1: “.. in my opinion, is needed that the authors demonstrated that the effects described on cell viability are selective or specific on cancer cell lines. What are the effects on non-cancer cell lines? If they are the same, the results have not validity for the last aim of the work.”

Authors reply: “The aim of this study was to test in a comprehensive manner a hypothesis which is generally accepted as a fact by the scientific community; that is that natural compounds can induce EMT in cancer cell lines and that this correlates with increased sensitivity to therapy. The significance of our findings is that this hypothesis is not correct at a general scale. And that if there is a relation between these two concepts it needs to be studied in much more detail. We have shown how such a study can be performed in another study of ours which has recently been published (Akbar et al. 2020, Journal of Cancer 11, 949).”

Second review comment. This comment has not been appropriately addressed by the authors. The authors do not demonstrated that the effects of different natural compounds used in these experimental conditions are selective or specific on cancer cell lines.

Authors reply: “We still do not understand why this question is relevant. What does the reviewer mean when he/she uses the terms “selective” and “specific”? If the question is: do these compounds affect normal (non-cancer) cells? then our response is: this question is irrelevant to this work as there are a number of papers where these compounds have been tested on cancer cells, as well as on normal cells (e.g., Breast-Fisetin-PMID: 26755433; Colon-EGCG- PMID: 29596305). The question we have is not whether they are also effective on normal cells. Our questions are (1) “do they cause MET in cancer; as has been argued in the literature”, and (2) “if they do cause MET, does this correlate with an increased sensitivity to chemotherapeutics; as has been argued in the literature”.

Third review comment: In my opinion, for a complete and correct interpretation of the results, the knowledge of the effects of different compounds on no-tumoral cells are needed.

  1. First review comment 2: “Pages 3 and 4, Tables 1-3. The authors described the effects of different compounds on cell viability of several cancer human cell lines (breast, colon, pancreatic). For having sure that the effects of each compound is specific or selective on the different cancerous cells, is needed to compare with the effect of each compound on a non-cancerous cell line (breast tissue, colon or intestinal cell; pancreatic cells). So, in each table, one or two new columns must be added in which the effects of each compound will be described. Only the compounds that have differential effects on the viability of both types of cell lines must be considered.”

Authors reply: “If the hypothesis that natural compound induced MET increases chemosensitivity had been validated here, we would have suggested the readers to use low doses of natural compounds (not toxic for any cell) and combine this with therapy. Since the hypothesis has not been validated, we have not planned additional experiments. There are several studies showing that natural compounds do not cause any damage to normal cells in low levels. (PMID: 15367704, PMID: 29034071, PMID: 31261976)

Second review comment. This aspect has not been appropriately addressed by the authors. The authors do not demonstrated that the effects of different natural compounds used were selective or specific on cancer cell lines. The indicated references are related only with three natural compounds (curcumin, apigenin, fisentin) on specific type of cells (hepatocytes, other, hepatocellular carcinoma cells, respectively). Nothing to see with the experiment described here.

Authors reply: “Please see our response to the first question.”

Third review comment: Please see my response to the first question.

  1. First review comment 3: “In my opinion, in the actual form, it is very difficult to follow and knowledge the results reported. Many compounds have been tested in many cell lines but so it is very difficult to extract some concrete conclusion. Authors must clarify what are the most important findings showed by this manuscript about the selective effects of the differents compounds tested.”

Authors reply: “Thank you for this comment. We have now included several summary tables into our manuscript.”

Second review comment. I had look for into the manuscript for the new summary tables and I had not found there. Tables and Supplementary Tables are the same that in the first version of the manuscript.

Authors reply: “Important findings are summarized in lines 126-132 and lines 166-171. Tables S1 and S3 are the summary tables and show all results.”

Third review comment: Done.

  1. First review comment 5: “Page 1, abstract, lines 15 and 17, sentences: “Natural products exhibit…” and “… effects of natural compounds…” These sentences are very imprecise and ambiguous. These sentences must be rewritte to concrete and avoid imprecise description.”

Authors reply: “As the abstract is only a general description of the study we have not changed the sentences.”

Second review comment: This aspect has not been appropriately addressed by the authors. What natural products exhibit antiproliferative activity against cancer cells? All of them? What natural compounds have been investigated? These sentences need to be clarify in the Abstract -the more read section of the manuscript- for avoid imprecise description.

Autors reply: “All compounds are anti-proliferative for at least one cell line, at least at one dose tested. We modified the abstract to include names of compounds tested and summary statements of our findings.”

Third review comment: Done.

  1. First review comment: “Page 4, lines 157-158. “An increase in CDH1, …and viceversa for EMT”. This affirmation must be justified appropriately”

Authors reply: “The sentence explains the choice of CHD1 and VIM expression levels as indicators of EMT and MET.”

Second review comment: This aspect has not been appropriately addressed by the authors. This affirmation must be appropriately explained and justified. The explanation and justification must be added to the manuscript because is central in the subsequent interpretations. In the actual version it is not justified.

Authors reply: “Transcriptomic based EMT phenotype scoring is reliable (Tan et al. 2014, EMBO Mol Med). Many studies have used CDH1 and VIM, two genes highly expressed in epithelial and mesenchymal cell states respectively, to characterize EMT status (Park et al. 2008, Genes&Dev, Qiu et al. 2020, RNA, Chakraborty et al. 2020, Front. Bioeng. Biotechnol). Evaluation of the CDH1/VIM gene pair has been shown to be a most reliable predictor of EMT for cell lines (George et al. 2017, Cancer Res). Our experience also shows that CDH1/VIM based evaluation of EMT is very reliable (Akbar et al. 2020, Journal of Cancer 11, 949).

We have added a paragraph to the discussion section summarizing these observations, and therefore why these two genes were selected in our work for the evaluation of EMT.”

Third review comment: Done.

  1. First review comment: “8. Statistical treatment of the results must be added into the tables and figures. Results must be expressed as mean ± standard error of the mean (for example) and the statistical significance for the different comparisons must be added.”

Authors reply: “The figures where Mann-Whitney U test were performed are now indicated.”

Second review comment: Authors change the statistical test applied but figures and tables continues practically as in the first version of the manuscript. Statistical treatment of the results has not been added to tables and figures.

Authors reply: “We have not “changed” the statistical test. We have only added the name of the test to the figures, as that is what we understood the reviewer requested. Statistical test were applied to Figure 1 and Figure S2 where asterisks indicate significant results. In Figure 2, Figure 3, Figure 4, Figure 5 and Table S2, the statistical test is the combination index (CI) scoring in which CI scores are indicated.”

Third review comment: In the Statistical Analysis subsection of the first version of the manuscript it say: “Statistical analysis was conducted by the GraphPad Prism v 6.01 and different treatment groups were compared using “t-test”. In the second version of the manuscript, in the same subsection it say: “Statistical analysis was performed using R Statistical Software and different treatment groups were compared using Mann-Whitney U test. Data from Tables 1-3 continuing been expressing only as the mean.

  1. First review comment: “9. The sections Results and Discussion must be unified as a only one “Results and Discussion” section. In the actual Results section it has been included explicative paragraph (for example: page 4, first paragraph of 2.2 subsection) no proper of this section.”

Authors reply: “The structure of the manuscript needs to follow the publisher’s guidelines. This is what we have done.”

Second review comment: This aspect has not been appropriately addressed by the authors.

Authors reply: We removed the introductory statement at the beginning of section 2.2.

Third review comment: Done.

  1. First review comment: “10. Page 12, Subsections 4.3-4.5. In all these subsections the number of independent repetitions (replicates) of the experiments for each treatment must be clearly indicated.”

Authors reply: “These have been clarified.”

Second review comment: I read 4.3-4.5 subsections and I had not found text added for clarification. The text are practically the same as in the first version of the manuscript.

Authors reply: This information has now been added to section 4.3 and 4.5.

Third review comment: Done.

Author Response

Third review comment: In my opinion, for a complete and correct interpretation of the results, the knowledge of the effects of different compounds on no-tumoral cells are needed.

Answer to 3rd round review comment: The table shown below lists at least one publication referring to the lack of cytotoxic effects on non-tumor cells. We have also added these references to our paper, together with a statement in the introduction. However, to perform all these experiments again in this paper, to us, is unnecessary.

Apigenin

PMID:28684964, PMID: 29034071

Curcumin

PMID: 15367704

EGCG

PMID: 29596305, PMID: 25839702

Fisetin

PMID: 31261976, PMID: 26755433

Forskolin

PMID: 26306624

Procyanidin B2

PMID: 25839702, PMID: 29027929

Resveratrol

PMID: 17785572, PMID: 30387805

Urolithin A

PMID: 33666815, PMID: 33158257

Fulvestrant

PMID: 23179494

Metformin

PMID: 22593441, PMID: 26725558

Third review comment: Please see my response to the first question.

Answer to 3rd round review comment: We have added additional references indicating that these compounds are not cytotoxic for non-tumor cells. No additional experiments are planned for this work.

Third review comment: In the Statistical Analysis subsection of the first version of the manuscript it say: “Statistical analysis was conducted by the GraphPad Prism v 6.01 and different treatment groups were compared using “t-test”. In the second version of the manuscript, in the same subsection it say: “Statistical analysis was performed using R Statistical Software and different treatment groups were compared using Mann-Whitney U test. Data from Tables 1-3 continuing been expressing only as the mean.

Answer to 3rd round review comment: We did use GraphPad Prism for generating the figures and we did use the same software to generate initial statistics. However, the final statistical results we generated using R. We have changed the depiction of data on Tables 1-3. They now include the standard error in addition to the mean.

Reviewer 2 Report

I have no more comments.

Author Response

There was no new comment

Reviewer 3 Report

No functional readout, the cell viability is not for EMT.

No key experiments are evaluated the statements of 

Author Response

No functional readout, the cell viability is not for EMT.

No key experiments are evaluated the statements that natural products affect EMT in cancer cell lines” by any migration and invasion experiments.

Cell viability is a measure of the proportion of live, healthy cells, does any literature say it for EMT experiments?

Answer to 3rd round review comment:  We are not evaluating cytotoxicity as a measure of EMT. We are trying to test if there is a relation between MET and increased cytotoxicity. The starting point of our work is that in many studies a relation between natural product induced MET and its correlation with increased sensitivity to chemotherapeutics was shown. Evaluation of EMT with marker genes CDH1 and VIM has been performed in innumerable studies. Phenotypic changes like migration and invasion result from these transcriptional changes. To add further experiments to this study is not justified given the fact that our results negate what has been stated in the literature. The novelty and importance of this study is exactly this; i.e. that it negates previous findings. Research in this field will benefit from our findings and will question the generality of previously conducted and published data.

Reviewer 4 Report

The authors did not eliminate significant serious flaws, in some cases they did not provide an adequate explanation. Research not conducted correctly. Research is overloaded with information, which interferes with the perception of it. In this form, the article is unlikely to be perceived by readers.

Author Response

The authors did not eliminate significant serious flaws, in some cases they did not provide an adequate explanation. Research not conducted correctly. Research is overloaded with information, which interferes with the perception of it. In this form, the article is unlikely to be perceived by readers.

Answer to 3rd round review comment: We did address all your comments above, as detailed and specific as we could be. If you have further specific questions or comments we would do our best to answer them.